# Combined Association of Plasma Metabolites with Body Mass Index and Physical Activity Level

**DOI:** 10.3390/biology13121074

**Published:** 2024-12-20

**Authors:** Mayara Lambert, Larissa de Castro Pedroso, Alex Aparecido Rosini Silva, Leonardo Henrique Dalcheco Messias, Andréia M. Porcari, Patrícia de Oliveira Carvalho, Pedro Paulo Menezes Scariot, Ivan Gustavo Masselli dos Reis

**Affiliations:** 1Research Group on Technology Applied to Exercise Physiology—GTAFE, Health Sciences Postgraduate Program, São Francisco University, Bragança Paulista 12916-900, SP, Brazil; lambert.mayara@mail.usf.edu.br (M.L.); larissa.pedroso@mail.usf.edu.br (L.d.C.P.); leonardo.messias@usf.edu.br (L.H.D.M.); pedro.scariot@mail.usf.edu.br (P.P.M.S.); 2MS4Life Laboratory of Mass Spectrometry, Health Sciences Postgraduate Program, São Francisco University, Bragança Paulista 12916-900, SP, Brazil; alex.rosini@mail.usf.edu.br (A.A.R.S.); andreia.porcari@usf.edu.br (A.M.P.); patricia.carvalho@usf.edu.br (P.d.O.C.)

**Keywords:** body mass index, index of physical activity questionnaire, obesity, metabolomic

## Abstract

**Simple Summary:**

This correlative and exploratory study aimed to comprehensively assess the plasma metabolite profiles of subjects with a lean versus overweight/obese body mass index, and low or high self-reported levels of physical activity using untargeted metabolomic and bioinformatic approaches. The majority of the changes detected in this study were attributable to body mass index, although some effect of self-reported levels of physical activity was also observed. Interaction effects were found when subgroups of body mass index were combined with subgroups of self-reported levels of physical activity, indicating a combined modulation.

**Abstract:**

Metabolomic analysis of the changes in plasma metabolites in obesity along with physical activity interaction may contribute to disease diagnosis and treatment. We sought to make a comprehensive assessment of the plasma metabolite profile of subjects with a lean (*n* = 20, BMI = 22.3) or overweight/obese (*n* = 29, BMI = 29) body mass index (BMI) and low (*n* = 33, IPAQ = 842) or high (*n* = 16, IPAQ = 6935) index of physical activity questionnaire (IPAQ), using an untargeted metabolomic approach. Two-way analysis of variance was applied to the data obtained from liquid chromatography–mass spectrometry analyses and resulted in 64 metabolites, mainly responsible for the data variance among the different groups. Finally, a complex network approach reveals the most relevant metabolites. The majority of the relevant metabolites are oxidized species of phospholipids. Most species of phosphatidylcholine and a species of phosphatidylglycerol were found to be decreased in obese subjects, while most species of phosphatidylethanolamine, phosphatidylserine, and phosphatidylinositol were increased. Only a single species each of prostaglandin, phosphatidylglycerol, and phosphatidylinositol were modulated by IPAQ, but interaction effects between BMI and IPAQ were found for most of the metabolites in the combination of obese BMI with low IPAQ.

## 1. Introduction

A threat to global health, obesity is a chronic disease caused by increased body fat deposition and is estimated to affect up to 30% of the world population, including overweight and obese individuals [1]. Obesity is typically diagnosed by estimating the body mass index (BMI), which is associated with total plasma levels of triglycerides, diglycerides, and free fatty acids [2], along with metabolic alterations incident to the disease [3]. Obesity is multifactorial, and the chance of developing this disease may be affected by endogenous factors like genetics [4], with an estimated heritability of 40–70% [5,6], and exogenous factors such as the environment and even socioeconomic status [7]. Due to the growth of the obesity pandemic in recent decades [8], obesity has been intensively studied in an attempt to better understand the etiology of the disease and to find more effective prophylaxis and therapy [9].

Metabolomics is the molecular study of a biological system with molar weight of up to 1500 Da [10]. There are thousand metabolites in the human organism and identifying them and discovering their roles still pose a challenge, even for the metabolomic approach. Several studies have reported metabolic patterns associated with obesity; liquid chromatography–mass spectrometry (LC–MS) characterization of the metabolomic profile [11] and the lipidomic signature of subjects with high levels of triglycerides and cholesterol [12] are recent advances in the study of obesity.

Compelling evidence shows that the physiological modifications caused by physical training can prevent or delay the onset of obesity [13], and that physical activity can modulate endogenous metabolites [14,15]. Therefore, obesity and the level of physical activity may play a role in modulating the same metabolites due to their close and antagonist relationship, but the metabolic alterations in the presence of the disease remain to be elucidated. Understanding and identifying the metabolomic fingerprint of the interaction between obesity and physical exercise may help to better understand human organism responses from a molecular perspective.

Changes in lipid metabolism associated to obesity have been linked to the development of obesity-related comorbidities [16,17]. These changes include elevated levels of plasma triglycerides, total cholesterol, low-density lipoprotein (LDL), oxidized LDL, and decreased concentrations of high-density lipoprotein (HDL) [18]. A variety of plasma metabolite families, including fatty amide, prenollipid, sphingolipid, branched-chain amino acid, and various derivatives of amino acids, acylcarnitines, fatty acids, and lysophospholipids, have been discovered to have associations with BMI [19,20,21].

Furthermore, changes in glycerophospholipids may contribute to the pathological process of metabolic diseases, according to a number of studies [19]. Numerous studies have reported phospholipids that characterize the obesity profile, but the direction of changes is controversial [3,18,22,23]. Obese subjects may exhibit increased levels of phosphatidylinositol (PI) [24]. Phosphatidylcholines (PCs) and phosphatidylethanolamines (PEs) account for 50% and 20–30% of the membrane composition, respectively [25], and disturbances in PCs and/or PEs may be a risk factor for obesity development [19,26]. Both choline and ethanolamine appear to be altered in the plasma/serum of obese subjects in other untargeted approaches. Increased lysophosphatidylcholine (lysoPC) C14:0 and lysoPC C18:0 were found in obese subjects [18], and several lysoPCs respond to fluctuations in BMI [21]. On the other hand, lysoPC C18:1 and C18:2 were decreased in obese children [23] and adults [18] and inversely correlated with BMI [27]. Other authors have reported decreased levels of these compounds [24,28,29] and correlation between plasma levels of lysoPC, sphingomyelins, and PCs with obesity after weight loss [30]. Nevertheless, whether plasma/serum levels of PCs and PEs may be related to obesity in humans still needs to be further investigated.

Regarding the isolated effect of physical activity, the literature is relatively scarcer. Associations for lysoPC(20:3), PC(16:0/20:3), PC(18:0/20:3), and PC(18:1/20:3) were inversely associated with concentrations of diabetes-associated glycerophospholipids [31]. However, when the level of physical activity was grouped as low or high (<2226 vs. ≥2226 MET-min/week in men; or <2079 vs. ≥2079 MET-min/week in women), the aforementioned associations among glycerophospholipids were observed only in participants with low physical activity [31].

In this exploratory study, the metabolomic fingerprints of Brazilian subjects with varying BMIs and levels of physical activity were identified. After assembling a network of correlations (Pearson), the main compounds were determined via an eigenvector centrality analysis.

## 2. Material and Methods

### 2.1. Ethics and Subjects

All procedures were approved by the Human Subject Research Ethics Committee of Sao Francisco University (Protocol number 12087719.5.0000.5514) before this study began. Twenty-seven female and twenty-two male subjects were enrolled in this study (Table 1) and signed a written informed consent form regarding the procedures involved in this investigation. The inclusion criteria were as follows: (I) body mass index above 18 kg/m^2^; (II) age between 18 and 60 years; (III) healthy values of body temperature, heart rate, and blood pressure; and (IV) fasting for 12 h. The exclusion criteria were as follows: (I) history of medical illness, drug, or alcohol abuse; (II) pregnancy or breastfeeding; (III) hospital admission or surgery within the last 3 months; (IV) blood donation or loss within the last month; (V) participation in another clinical trial within the last 3 months; and (VI) chronic medication use or use within the last 2 weeks.

### 2.2. Determination of the Body Mass Index

Anthropometric data of weight and height were obtained with a scale and stadiometer (Welmy, Sao Paulo, Brazil). The BMI was calculated as the ratio of body weight (kg) to height squared (m^2^). Values less than 25 kg/m^2^ were classified as lean, and values greater than 25 kg/m^2^ were classified as overweigh/obese.

### 2.3. Determination of the Level of Physical Activity (Index of Physical Activity Questionnaire–IPAQ)

The subjects answered 6 topics from the short version of the Index of Physical Activity Questionnaire (IPAQ), validated to the Brazilian population, regarding their physical activity frequency, duration, and intensity, to assess their physical activity level [32,33]. Participants were classified into two levels of physical activity based on the IPAQ: low activity (IPAQ ≤ 3066) and high activity (IPAQ > 3066).

### 2.4. Experimental Design

#### 2.4.1. Grouping the Subjects by the BODY MASS INDEX

In one of the two approaches, the overall sample was segmented into two groups based on objective criteria related to BMI (Lean: 18 < BMI < 25; Overweight/Obese: BMI ≥ 25) to search for differences in lipid metabolites between these groups (Table 1).

#### 2.4.2. Grouping the Subjects by the Physical Activity Level (IPAQ)

In the other approach, the same sample was segmented into two groups based on objective criteria related to the IPAQ (Low: IPAQ ≤ 3066; High: IPAQ > 3066) to search for lipid metabolites modulated by the level of physical activity (Table 1).

#### 2.4.3. BMI/IPAQ Interactions

In addition to the aforementioned groups, statistical differences were verified for the following combinations (interactions) between subgroups of BMI and IPAQ: Lean + Low (*n* = 13); Lean + High (*n* = 7); Overweight/Obese + Low (*n* = 20); Overweight/Obese + High (*n* = 9).

### 2.5. Metabolomics Analysis

Intravenous blood samples (5 mL) were obtained from the forearm by a trained professional into centrifuge tubes containing EDTA anticoagulant. Blood was drawn from the subject in a sitting position in the morning after a 12 h fast. The samples were centrifuged at 1500 rpm for 10 min, and the plasma fractions were separated and stored in microcentrifuge tubes at −80 °C.

A pooled sample was formed before sample extraction by combining equal parts of each sample (20 μL), which were then aliquoted into different quality control (QC) samples and extracted alongside the other samples. Plasma samples (150 μL) were randomized and mixed with cold isopropanol (200 μL), vortexed for 30 s, and then centrifuged (12,000 rpm, 4 °C, 10 min). Subsequently, the supernatant (200 μL) was collected and dried under N2. Blank samples were prepared using ultra-pure water instead of plasma. To monitor deviations in extraction and system stability, QC samples were inserted after every 10 samples. Additionally, a QC sample was employed at the outset of the experiment to facilitate instrumental stabilization of the LC–MS system. Participant samples were extracted and analyzed in a randomized manner to observe biological variation and minimize instrumental bias.

The analysis was adapted from Silva et al. [34]. An ACQUITY UPLC was used, coupled to a XEVO-G2XS (QTOF) quadruple time-of-flight mass spectrometer (Waters, Manchester, UK) equipped with an ESI (Electrospray Ionization) source. For lipidomics analysis, an ACQUITY UPLC^®^ CSH C18 column (2.1 mm × 100 mm × 1.7 μm, Waters, Manchester, UK) was employed, using mobile phase A composed of an acetonitrile (ACN):water (H2O) solution (60:40, *v*/*v*) with 10 mM ammonium formate + 0.1% formic acid, and mobile phase B composed of isopropanol:acetonitrile (ACN) (90:10, *v*/*v*) with 10 mM ammonium formate + 0.1% formic acid, with a flow rate of 0.4 mL min^−1^.

The gradient started at 40% B, increasing to 43% within 2.0 min, further rising to 50% over 0.1 min, then reaching 54% within the next 9.9 min. Subsequent increments led to 70% B over 0.1 min, 99% B over 5.9 min, and eventually returning to 40% B over 0.1 min for column re-equilibration over the next 1.9 min. The total run time was 20 min. Data were recorded separately in positive (+) and negative (−) ion modes across the 50–1700 m/z range with an acquisition time of 0.5 s per scan. Source and desolvation temperatures were set at 140 °C and 550 °C (+), 400 °C (−), respectively, with a desolvation gas flow of 900 L h^−1^. Capillary voltages were 3.0 kV (+) and 2.5 kV (−), with a cone voltage of 40 V. S Leucine enkephalin (molecular weight = 555.62; 200 pg μL^−1^ in 1:1 ACN:H2O) served as a lock mass for precise mass measurement [12].

### 2.6. Statistical Analysis

For didactical reasons, the statistical section was divided into different steps: data processing (Section 2.6.1), data analyses and metabolite selection (Section 2.6.2), and metabolite selection and complex network analysis (Section 2.6.3). These steps are illustrated in Figure 1.

#### 2.6.1. Data Processing

LC–MS raw files were processed using Progenesis QI 2.4 software (Nonlinear Dynamics, Newcastle, UK) for peak alignment, deconvolution, selection of possible adducts, and compound annotation based on data-independent acquisition (MSE). Metabolite identification relied on MS1 and MS2 experiments [35]. In the same spectrum, both low- and high-energy acquisition provided information on precursor ions (mass error ≤ 5 ppm) and fragments (mass error ≤ 10 ppm). Fragmentation Score, Mass Accuracy, Mass Error, Isotope Similarity, and Physiological function were assessed to accept the annotated molecules. To ensure compatibility between Progenesis PQI data and external SDF-based spectra libraries, we developed an in-house software called “SDF2PQI” to enhance fragment matches: https://github.com/pedrohgodoys/sdf_to_pqi/ (accessed on 11 July 2024) [36]. External SDF-based spectra libraries utilized included LipidMaps: http://www.lipidmaps.org/ (accessed on 11 July 2024), Human Metabolome Database: http://www.hmdb.ca/metabolites (accessed on 11 July 2024), and MoNA–MassBank of North America: https://mona.fiehnlab.ucdavis.edu/ (accessed on 11 July 2024). The identified levels were based on Sah et al. [37] and Liebicsh et al. [38] (Appendix A).

Data processing was performed with Metaboanalyst 6.0 (Xia Lab). Analysis of the LC–MS revealed 2534 compounds with different mass/charge ratios. Among these, 719 were removed based on quality control relative standard deviation values, resulting in 1815 features normalized to the sample median, transformed with the cube root, and scaled by range (Figure 1A).

#### 2.6.2. Data Analysis and Metabolite Selection

A two-way analysis of variance (Metaboanalyst 6.0, Xia Lab: https://metaboanalyst.ca/MetaboAnalyst/home.xhtml, accessed on 11 July 2024) was applied to identify the main lipids contributing to the variance in the data across groups based on BMI or IPAQ levels. Features with a significant difference (raw *p* < 0.05) were selected in the two-way ANOVA between the groups of BMI or IPAQ or their combination (Figure 1B1).

The relevant metabolites for BMI, IPAQ, or their interaction were combined into a single set comprising 64 metabolites (Figure 1B2) for the assembly of the correlation network.

#### 2.6.3. Complex Network Analysis

Based on significant Pearson correlations (*p* < 0.05) between the 64 relevant metabolites, a complex network topology was constructed (Figure 1C1). Both positive and negative correlations were bidirectional and equally significant in this case. The correlation coefficient between the linked nodes (metabolites), which varies from 0.01 to 1 (negative coefficients are treated as positive, and greater values indicate a stronger correlation), and the proximity degree to the BMI node, which ranges from 0.01 to 1 (higher values indicate closer proximity), are multiplied to determine the edge weight. Stated differently, the edge weight is calculated by dividing the correlation coefficient between two nodes by the least number of edges needed to reach the BMI node. Therefore, when edges were directly linked (correlated) to the BMI, they were allocated a weight equal to their respective correlation coefficient. Second-degree connections with the BMI were assigned a weight of 0.5 (half) of the correlation coefficient, while third, fourth, and fifth-degree connections were assigned weights of 0.25, 0.125, and 0.0625, respectively [39]. The target eigenvector scores were computed using these edge weights as connection strengths. Using the NetworkX 2.5 module for the Python programming language in the Jupyter Notebook integrated development environment, eigenvector centrality scores were calculated. The i^th^ element of the vector x formed with the equation Ax = λx, where A is the adjacency matrix of the graph G with eigenvalue λ, represents the eigenvector centrality for node I. If λ is the biggest eigen-value of the adjacency matrix A, then there exists a unique solution x for which all entries are positive. A node’s centrality is calculated with the targeted eigenvector using the weights of its edge connections and the centrality of its neighbors [40] (Figure 1C2).

## 3. Results

There is no association between BMI and IPAQ (−0.2, *p* = 0.16,) when analyzing the overall sample (*n* = 49). However, different patterns of association were found between subgroups of lean or overweight/obese BMI and low or high IPAQ. Moreover, the only significant correlation found was the direct association between overweight/obese BMI and high IPAQ (Table 2).

From a total of 1815 molecular features, 64 metabolites were found to have altered concentrations between groups (Appendix A). As shown in the Venn diagram (Figure 2), these significant differences were mainly due to BMI rather than IPAQ. When separated by groups, exactly 64 metabolites were shown to be affected by BMI, 5 by IPAQ, and 26 by interactions between BMI and IPAQ phenotypes (Lean + Low or Lean + High or Overweight/Obese + Low or Overweight/Obese + High). Interaction effects occur when the combined influence of factors affects the dependent measure. In the presence of an interaction effect, the effect of one factor is contingent upon the level of the other factor. The intersection between BMI and IPAQ (purple area) comprises 3 metabolites, while the intersection between BMI and Interaction (orange area) includes 24 metabolites, and the intersection between IPAQ and Interaction (green area) contains no metabolites. Only two metabolites are present in the intersection among BMI, IPAQ, and Interaction (brown area).

A network was assembled based on significant correlations among the 64 upregulated metabolites, and eigenvector centrality was computed (Figure 3). Eigenvector centrality indicates the relevance of the metabolite to the network system and is based on the relevance of its neighbors. Initially, a weight value based on the coefficient of correlation between two metabolites and their association with BMI was attributed to each edge of the network (see methods for more details). Then, a first eigenvector value for each node is calculated by the sum of its links. The calculation of the eigenvector is repeated until the values of the network have become stable.

Regarding the statistical main effects, all identified metabolites present in the correlation network were significantly different between BMI groups, while few were different between IPAQ groups (Table 3). Not-identified compounds can be found in the Appendix A.

The metabolites with the 25 highest eigenvector centralities were selected for further discussion and are shown in a bar graph (Figure 4). The chemical class of the metabolites in the top 25 eigenvector centralities mainly comprised oxidized compounds of phosphatidylcholine (PC), phosphatidylethanolamine (PE), phosphatidylserine (PS), phosphatidylglycerol (PG), and phosphatidylinositol (PI), alongside non-oxidized PE and prostanoid (prostaglandin). Among the 25 highest eigenvector centrality scores, six metabolites were exclusively modulated by BMI (red). Additionally, 3 metabolites were simultaneously influenced by both BMI and IPAQ (purple), while 15 metabolites were simultaneously modulated by BMI and by an BMI/IPAQ interaction (orange). Only two metabolites were modulated simultaneously by BMI, IPAQ, and Interaction (brown) (Figure 4).

Both BMI and the interaction between overweight/obese BMI and low IPAQ promoted increased abundance of the same subclasses: PC(18:1), 2 PE(20:3), 2 PE(20:2/PGE1), Prostaglandin D1, PS(18:0/20:4), and 2 PS(22:4/PGE1), indicating a cumulative effect between factors (Table 4). Another evidence of a cumulative effect between factors, IPAQ along with the interaction between lean BMI and high IPAQ, promoted increased abundance of PG(40:5) (Table 4). Moreover, all changes found for the interaction between lean BMI and high IPAQ were in the opposite direction of the changes found for obese/overweight BMI (Table 4).

## 4. Discussion

The lack of correlation between BMI and IPAQ (−0.2, *p* = 0.16) is coherent with the heterogeneity of the sample, which included subjects with lean or overweight/obese BMI and low or high IPAQ. Instead of an indirect association between BMI and IPAQ, an unexpected direct correlation between overweight/obese BMI and high IPAQ was found. However, several metabolites were significantly associated in opposite ways with both BMI and IPAQ (Appendix A). These pieces of evidence reinforce the importance of studying the interaction between subgroups of both BMI and IPAQ. To the best of our knowledge, this is the first molecular evidence of the combined relationship of BMI and IPAQ with plasma metabolites.

The association of oxidized plasma compounds with obesity was not the scope of this work, but almost all of the metabolites discussed here were oxidized species of phospholipids. Adipocyte hypertrophy induces endoplasmic reticulum dysfunction and the consequent accumulation of reactive oxygen species [41], leading to the establishment of low-level and chronic inflammation in obese subjects [42]. In this scenario, immunological disorders may occur as a consequence of the activation of both cell inflammatory signaling cascades [43] and the innate immune system [42]. In fact, inflammatory factors have been suggested as biomarkers of obesity [44] as well as the altered expression of genes related to metabolism and the production of adipokines [45,46,47]. Therefore, the decrease of eight PCs, two PEs, and a PG in the overweight/obese group could be due to an increase in oxidative stress and degradation via lipid peroxidation processes that can occur in the disease. Together, our findings and the body of evidence indicate the relevance of alterations in oxidative metabolism during the disease process.

### 4.1. Phosphatidylcholines and Phosphatidylethanolamines

Unsurprisingly, most of the 25 metabolites highlighted via the network model are PCs and PEs, which are the major glycerophospholipids that make up cellular membranes. In this context, it has been demonstrated that obese mice treated with PC 18:0/18:1 show increased glucose tolerance and insulin sensitivity [48], and evidence from other animal studies suggests a role of the PC:PE ratio in regulating muscle metabolism [49]. Additionally, PC modulates fatty acids, phospholipids, and triacylglycerol synthesis, as well as their concentrations in plasma, through changes in the master regulator of de novo synthesis [25]. Increases in lysoPC have been suggested to be related to a decrease in catabolism due to increased caloric intake [21]. In the present findings, the decreased abundance of six out of seven PCs and the increased abundance of six out of eight PEs may be evidence of a reduced PC:PE ratio in the overweight/obese group (Figure 5).

Regarding IPAQ, there was no difference between groups (Figure 6). Instead, IPAQ showed significant correlations primarily with PCs and PEs (Appendix A). Among these correlations, three out of four associations found between PCs and IPAQ were positive, while all three significant associations between PEs and IPAQ were negative. Furthermore, the same PCs and PEs were associated with BMI in opposite directions.

In summary, these new insights into the relationship between BMI and IPAQ indicate that both obesity and low levels of physical activity may reduce plasma PCs and increase PE abundances.

Higher levels of physical activity are known to prevent the initial phase of the atherosclerotic process by enhancing HDL-mediated cholesterol efflux from arterial endothelium [50]. Despite PCs being present in HDL, high IPAQ scores did not improve lipid profiles by increasing PC levels (Figure 6). Instead, predominantly decreased PC levels were observed in individuals with obese/overweight BMI, accompanied by increases in PEs (Figure 5).

### 4.2. Phosphatidylglycerol

Special attention would be directed to PG, as it was the only plasma compound shown to be affected by both BMI and IPAQ independently, as well as by the interaction between overweight/obese BMI and low IPAQ (Figure 5 and Figure 6). PG levels are directly associated with adiposity and BMI, which corroborates PG’s function in preserving adipose tissue by inhibiting lipolysis under conditions of catabolic inflammation [51]. Despite this, the results of plasmatic LC–MS indicate a reduction in the species PG 40:5;O2 in subjects with overweight/obese BMI and an increase in those with high IPAQ scores. It is worth mentioning that different species of the same phospholipid class can exhibit different patterns in metabolic disease conditions. A decrease in the species PG 36:2 was observed in the liver under a condition commonly associated with obesity, albeit without changing the total levels of PG [52]. Therefore, plasmatic PG 40:5;O2 may be a metabolite sensitive to metabolic alterations due to obesity and physical activity levels, which appear to affect it in opposite directions and in a linear manner (significant correlations, Appendix A).

### 4.3. Phosphatidylinositol

Intermediates of PI play key roles in integrating the insulin-stimulated receptor with the cell membrane translocation of the glucose transporter 4 [53] and are involved in the development of obesity [54], particularly phosphatidylinositol-3,4,5-trisphosphate, which activates protein kinase B [55]. Additionally, the increased phosphorylation of PI species by phosphoinositide 3-kinase may play a role in the development of obesity-induced inflammation and insulin resistance [56]. On the other hand, the decrease in PI levels in the high-IPAQ group may be associated with the inhibition of phosphoinositide 3-kinase, which has been shown to protect against high-fat diet-induced insulin resistance in mice [56]. However, impaired phosphoinositide 3-kinase signaling has been demonstrated to cause peripheral insulin resistance [57]. 

The decrease in the high-IPAQ PI levels may be associated with a reduced emanation as triacylglycerol from adipocyte and/or an upregulation of the phosphatidic acid synthesis by the pathway that utilizes diacylglycerol originated by inositol removal of PI [58]. This is supported by the phosphatidic acid role in the activation of the target of rapamycin complex 1 that has been suggested to mediate the resistance exercise-stimulated protein synthesis [59]. Although obesity may be related to plasma levels of PI, the conclusions derived are speculative.

### 4.4. Prostaglandin

A novelty is the result of an increased abundance of plasma prostaglandin D1 (PGD1) in the obese group, while prostaglandin D2 (PGD2) was increased in the high-IPAQ group (Figure 5 and Figure 6). Prostaglandins are oxygenated fatty acids derived from arachidonic acid produced by the interaction between cytosolic phospholipase and cellular membranes [60]. PGD2 is associated with the regulation of various physiological processes and the development of diseases [60]. Previous studies on PGD2 have focused on its effects on sleep control, pain, appetite, inflammation, hypertension, cardiovascular diseases, diabetes, and obesity, but few studies have reported the levels of PGD2 in disease patients. Less studied than PGD2, PGD1 has been shown to be induced by inflammatory stimuli of atopic dermatitis and to have anti-inflammatory effects [61]. Moreover, PGD1 has been shown to reduce vascular permeability induced by other prostaglandins [62]. Therefore, further conclusions or even speculations about the role of PGD1 and PGD2 in obese and physically active subjects will require more in-depth investigation.

### 4.5. Phosphatidylserine

Another novelty was the increased abundance of two species of PS in the obese group (Figure 5). PS, a phospholipid of the inner layer of the cell membrane, is involved in the signaling and function of intracellular proteins [63]. Additionally, translocation of oxidized PS to the outer layer of the cell surface is a signaling mechanism for cell apoptosis [45], and oxidized PS is reported to have both anti- and pro-inflammatory effects [64,65]. Thus, the identification of oxidized PS in biological samples has the potential to serve as an obesity-specific biomarker for metabolic alterations.

### 4.6. Limitations and Future Directions

The limitations of the present experimental design include the indirect diagnosis of obesity (BMI), indirect estimation of the level of physical activity (IPAQ), and the lack of validation through biomarker measurements. Another limitation is the lack of control for age and sex, as both may influence obesity outcomes [66,67]. Although the correlations between BMI and biomarkers generally decrease with increasing age, this pattern is also observed with technically advanced measures for obesity diagnosis [67]. Therefore, measurement errors and/or recall bias could not be avoided. Future investigations can improve this aspect by restricting the sample to specific age and sex groups, including the evaluation of biomarkers frequently associated with obesity, and assessing parameters of physical performance such as aerobic and anaerobic capacities. Despite these limitations, this study represents a first step in understanding whether metabolites can be influenced by physical condition (BMI and physical activity level). Another strength of this study is being the first to integrate and explore two scientific strategies (metabolomics and complex network analysis). The combination of these two strategies is of interest to physiologists, but its application is still limited, especially using BMI and IPAQ data. This may have interesting practical applications in medical, nutritional, and biological experimental designs.

## 5. Conclusions

The majority of relevant metabolites are oxidized species of phospholipids. Most species of PCs and a PG species were found to be decreased in obese subjects, whereas most species of PEs, PSs, and a PI were increased. Only one species each of PGD, PG, and PI were altered with IPAQ, but interaction effects between BMI and IPAQ were found for most metabolites in the combination of obese BMI with low IPAQ. A notable finding was the increased abundance of plasma PGD1 in obese subjects, while PGD2 levels were higher in the high-IPAQ group. Another significant result was the increased abundance of two species of PS in the obese group. PG deserves special attention as it was the only plasma compound affected by both BMI and IPAQ independently, as well as by the interaction between overweight/obese BMI and low IPAQ. This may provide molecular evidence of the combined relationship of BMI and IPAQ with plasma metabolites.

## Figures and Tables

**Figure 1 biology-13-01074-f001:**
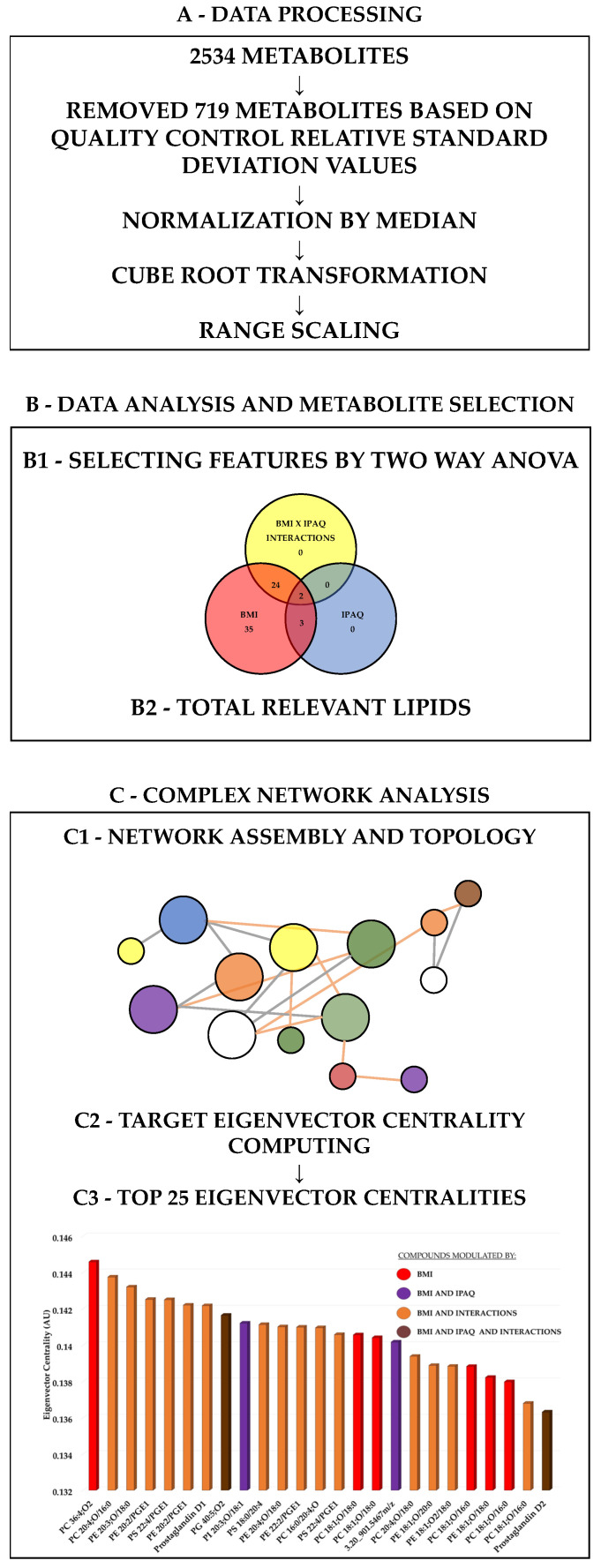
Sequence of data processing and analysis sequence: (**A**) data processing; (**B**) data analysis and metabolite selection; and (**C**) complex network analysis.

**Figure 2 biology-13-01074-f002:**
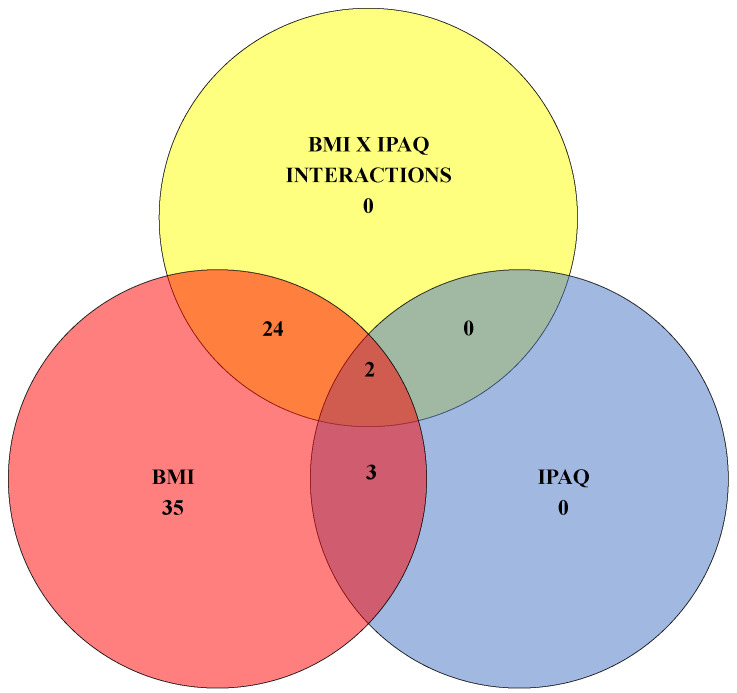
Venn diagram illustrating the number of metabolites modulated by the body mass index (BMI) in regions colored red, orange, purple, and brown; the number of metabolites influenced by the index of physical activity (IPAQ) in regions colored blue, purple, green, and brown; and the number of metabolites affected by the BMI/IPAQ interaction in regions with yellow, orange, green, and brown colors.

**Figure 3 biology-13-01074-f003:**
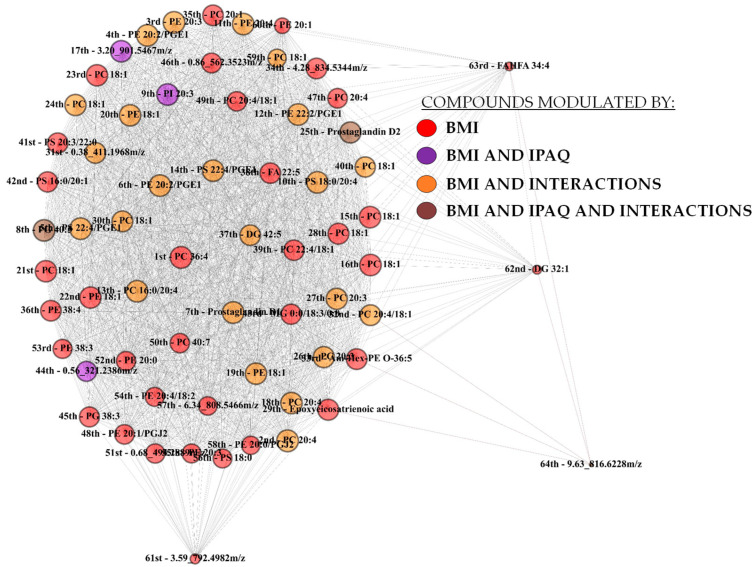
Representation of the complex network assembled with 64 metabolites as nodes and significant correlations between them as edges. The size of the node is equivalent to its eigenvector or relevance in the network. Metabolites modulated by the body mass index are shown in colors red, purple, orange, and brown, while those modulated by the index of physical activity questionnaire are represented in colors purple and brown. Metabolites affected by an interaction between BMI and IPAQ are depicted in colors orange and brown. Negative and positive correlations are represented by brown and gray edges, respectively.

**Figure 4 biology-13-01074-f004:**
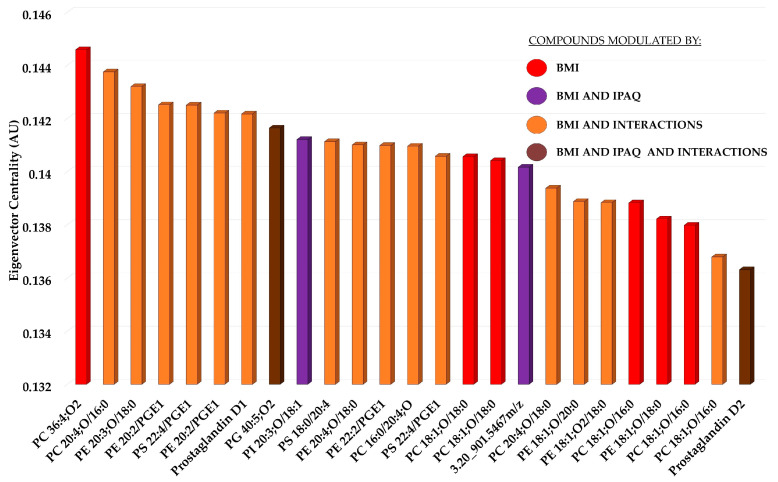
Top 25 highest BMI eigenvector centralities from the correlation network of metabolites modulated by the body mass index (red, purple, orange, and brown), the level of physical activity (purple and brown), or their interaction (orange and brown).

**Figure 5 biology-13-01074-f005:**
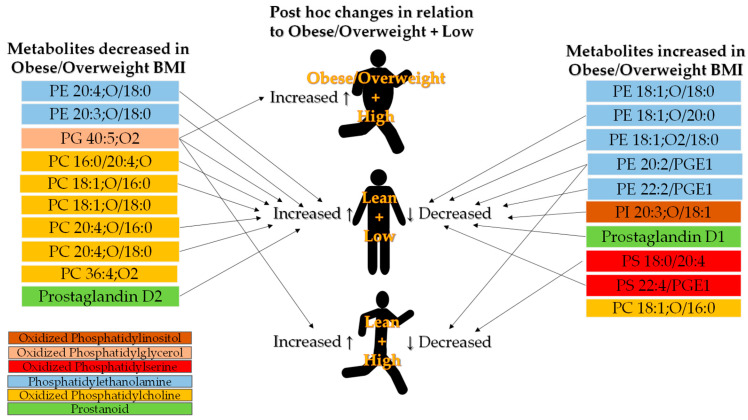
Depiction of results of decreased (**left**) and increased (**right**) metabolites in subjects with obese/overweight BMI, along with post hoc changes in relation to the obese/overweight + low group (**center**).

**Figure 6 biology-13-01074-f006:**
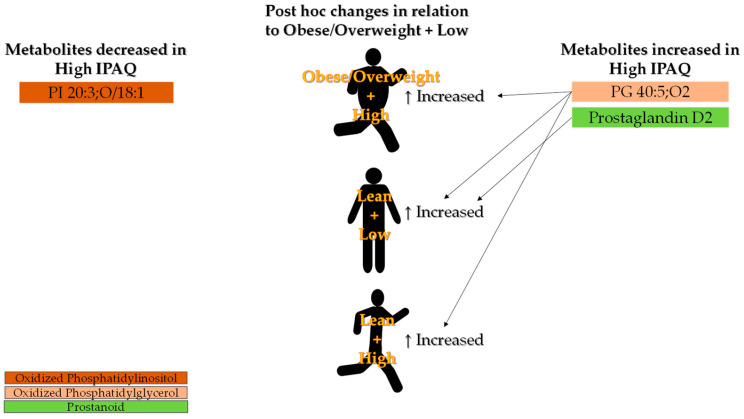
Depiction of results showing decreased (**left**) and increased (**right**) metabolites in subjects with high IPAQ, along with post hoc changes relative to the obese/overweight + low group (**center**).

**Table 1 biology-13-01074-t001:** Descriptive statistics when grouping the subjects by the body characteristics (according to BMI) and physical activity levels (according to IPAQ).

	Grouping by the BMI		Grouping by the IPAQ	
	Lean 18 < BMI ˂ 25	Overweight/Obese BMI ≥ 25	Low IPAQ ≤ 3066	High IPAQ > 3066
Sex	F = 10, M = 10	F = 17, M = 12	F = 19, M = 14	F = 8, M = 8
Age (years)	29 ± 10	35 ± 13	34 ± 14	29 ± 9
Body Mass (kg)	62.6 ± 8	78.3 ± 10 #	73.1 ± 13	69.4 ± 9
Height (m)	1.67 ± 0.1	1.64 ± 0.1	1.66 ± 0.1	1.66 ± 0.1
BMI (kg/m^2^)	22.3 ± 2	28.9 ± 3 #	26.8 ± 5	25.2 ± 2
IPAQ (Score)	3361 ± 2823	2467 ± 4019	842 ± 890	6935 ± 3538 *

Data are in the mean ± SD. One-way ANOVA was used (Statisca 7.0–Statsoft). # indicates significant differences (*p* < 0.05) in relation to lean group. * indicates significant difference (*p* < 0.05) in relation to Low-IPAQ group. F = Female and M = Male.

**Table 2 biology-13-01074-t002:** Correlation coefficients between subgroups of lean or overweight/obese body mass index (BMI) with low or high physical activity questionnaire (IPAQ).

	Lean BMI	Overweight/Obese BMI
Low IPAQ	0.29 (*n* = 13)	−0.42 (*n* = 20)
High IPAQ	−0.28 (*n* = 7)	0.71 * (*n* = 9)

* Significant correlation (*p* < 0.05).

**Table 3 biology-13-01074-t003:** Eigenvector rank, chemical class, metabolite description, adducts, formula, and main effects values (*p*) of all identified metabolites present in the correlation network.

Eigenvector Rank	Chemical Class	Metabolite	Adducts	Formula	BMI (*p*)	IPAQ (*p*)
1st	Oxidized Phosphatidylcholine	PC 36:4;O2	M+FA-H	C44H80NO9P	0.009	0.098
2nd	Oxidized Phosphatidylcholine	PC 20:4;O/16:0	M+FA-H	C44H80NO9P	0.009	0.116
3rd	Phosphatidylethanolamine	PE 20:3;O/18:0	M-H2O-H. M-H	C43H78NO9P	0.004	0.133
4th	Phosphatidylethanolamine	PE 20:2/PGE1	M+FA-H	C45H80NO11P	0.010	0.179
5th	Oxidized Phosphatidylserine	PS 22:4/PGE1	M+FA-H	C48H80NO13P	0.004	0.199
6th	Phosphatidylethanolamine	PE 20:2/PGE1	M+FA-H	C45H80NO11P	0.008	0.220
7th	Prostanoid	Prostaglandin D1	M-H2O-H	C20H34O5	0.003	0.128
8th	Oxidized Phosphatidylglycerol	PG 40:5;O2	M+FA-H	C46H81O12P	0.002	0.031
9th	Oxidized Phosphatidylinositol	PI 20:3;O/18:1	M-H	C47H83O14P	0.003	0.036
10th	Oxidized Phosphatidylserine	PS 18:0/20:4	M-H	C45H80NO9P	0.005	0.171
11th	Phosphatidylethanolamine	PE 20:4;O/18:0	M-H	C43H78NO9P	0.008	0.135
12th	Phosphatidylethanolamine	PE 22:2/PGE1	M+FA-H	C47H84NO11P	0.004	0.141
13th	Oxidized Phosphatidylcholine	PC 16:0/20:4;O	M+FA-H	C44H80NO9P	0.011	0.133
14th	Oxidized Phosphatidylserine	PS 22:4/PGE1	M+FA-H	C48H80NO13P	0.008	0.142
15th	Oxidized Phosphatidylcholine	PC 18:1;O/18:0	M+FA-H	C44H84NO9P	0.004	0.214
16th	Oxidized Phosphatidylcholine	PC 18:1;O/18:0	M+FA-H	C44H84NO9P	0.004	0.184
18th	Oxidized Phosphatidylcholine	PC 20:4;O/18:0	M+FA-H	C46H84NO9P	0.005	0.134
19th	Oxidized Phosphatidylethanolamine	PE 18:1;O/20:0	M-H	C43H82NO9P	0.002	0.201
20th	Oxidized Phosphatidylethanolamine	PE 18:1;O2/18:0	M-H2O-H	C41H80NO10P	0.004	0.191
21st	Oxidized Phosphatidylcholine	PC 18:1;O/16:0	M+FA-H	C42H80NO9P	0.014	0.229
22nd	Oxidized Phosphatidylethanolamine	PE 18:1;O/18:0	M-H2O-H	C41H80NO10P	0.006	0.146
23rd	Oxidized Phosphatidylcholine	PC 18:1;O/16:0	M+FA-H	C42H80NO9P	0.004	0.164
24th	Oxidized Phosphatidylcholine	PC 18:1;O/16:0	M+FA-H	C42H80NO9P	0.005	0.186
25th	Prostanoid	Prostaglandin D2	M-H	C20H32O5	0.001	0.022
26th	Oxidized Phosphatidylcholine	PC 20:3;O2/18:1	M+FA-H	C46H84NO9P	0.004	0.138
27th	Oxidized Phosphatidylcholine	PC 20:3;O/18:0	M+FA-H	C46H84NO9P	0.005	0.111
28th	Oxidized Phosphatidylcholine	PC 18:1;O/20:4	M+FA-H	C46H80NO9P	0.004	0.091
29th	Epoxyeicosatrienoic acid	Epoxyeicosatrienoic acid	M-H	C20H32O3	0.011	0.078
30th	Oxidized Phosphatidylcholine	PC 18:1;O2/18:3	M+FA-H	C44H80NO10P	0.014	0.159
32nd	Oxidized Phosphatidylcholine	PC 20:4/18:1;O	M+FA-H	C46H80NO9P	0.005	0.124
33rd	Glycerophosphoethanolamine glycans	Am-Hex-PE O-36:5	M-H2O-H	C47H84NO12P	0.002	0.111
35th	Oxidized Phosphatidylcholine	PC 20:1;O	M-H2O-H	C28H54NO9P	0.000	0.270
36th	Oxidized Phosphatidylethanolamine	PE 38:4;O	M-H	C43H78NO9P	0.001	0.136
37th	Diacylglycerol	DG 42:5;O2	M+Na-2H	C45H78O8	0.000	0.086
38th	Docosanoid	FA 22:5;O2	M-H2O-H	C22H34O4	0.006	0.068
39th	Oxidized Phosphatidylcholine	PC 22:4/18:1;O	M+FA-H	C48H84NO9P	0.002	0.072
40th	Oxidized Phosphatidylcholine	PC 18:1;O/18:2	M+FA-H	C44H80NO9P	0.006	0.155
41st	Phosphatidylserine	PS 20:3/22:0	M-H2O-H	C48H88NO10P	0.002	0.087
42nd	Diacylglycerophosphoserine	PS 16:0/20:1	M+FA-H	C42H80NO10P	0.004	0.246
43rd	Monoacylglycerol	MG 0:0/18:3/0:0	2M-H	C21H36O4	0.001	0.293
45th	Oxidized Phosphatidylglycerol	PG 38:3;O2	M+FA-H	C44H81O12P	0.001	0.416
47th	Oxidized Phosphatidylcholine	PC 20:4;O/20:0	M-H2O-H	C48H88NO9P	0.002	0.123
48th	Oxidized Phosphatidylethanolamine	PE 20:1/PGJ2	M-H2O-H	C45H78NO10P	0.006	0.385
49th	Phosphatidylcholine	PC 20:4/18:1	M-H	C46H82NO8P	0.003	0.352
50th	Oxidized Phosphatidylcholine	PC 40:7;O	M-H	C48H82NO9P	0.004	0.149
52nd	Phosphatidylethanolamine	PE 20:0	M-H	C25H50NO8P	0.008	0.112
53rd	Oxidized Phosphatidylethanolamine	PE 38:3;O	M-H	C43H80NO9P	0.012	0.166
54th	Phosphatidylethanolamine	PE 20:4/18:2	M-H. M+Na-2H	C43H74NO8P	0.002	0.363
55th	Oxidized Phosphatidylethanolamine	PE 20:3;O/16:0	M-H	C41H76NO9P	0.007	0.073
56th	Acylglycerophosphoserine	PS 18:0;O/20:0	M+Na-2H	C44H88NO9P	0.005	0.316
58th	Oxidized Phosphatidylethanolamine	PE 20:0/PGJ2	M-H2O-H	C45H80NO10P	0.001	0.108
59th	Oxidized Phosphatidylcholine	PC 18:1;O/16:1	M-H2O-H	C42H78NO9P	0.002	0.582
60th	Oxidized Phosphatidylethanolamine	PE 20:1;O	M-H2O-H	C25H48NO9P	0.015	0.648
62nd	Diacylglycerol	DG 32:1	M+FA-H	C35H66O5	0.004	0.266
63rd	Fatty Acyl esters of Hydroxy Fatty Acids	FAHFA 34:4	M-H	C34H58O4	0.008	0.758

Statistical significance is displayed in red (*p* < 0.05).

**Table 4 biology-13-01074-t004:** Direction of abundance changes in plasma metabolites and post hoc interactions in the top 25 eigenvector rank of centralities.

		BMI		IPAQ		Post Hoc Interactions				
Eigenvector Rank	Metabolite	Obese	*p*	High	*p*	Lean + Low	Lean + High	Obese + Low	Obese + High	*p*
4th	PE 20:2/PGE1	↑	0.010	-	0.179	↓ *	↓ *	↑	-	0.040
5th	PS 22:4/PGE1	↑	0.004	-	0.199	↓ *	-	↑	-	0.047
6th	PE 20:2/PGE1	↑	0.008	-	0.220	↓ *	↓ *	↑	-	0.030
7th	Prostaglandin D1	↑	0.003	-	0.128	↓ *	-	↑	-	0.038
9th	PI 20:3;O/18:1	↑	0.003	↓	0.036	-	-	-	-	0.076
10th	PS 18:0/20:4	↑	0.005	-	0.171	-	↓ *	↑	-	0.033
12th	PE 22:2/PGE1	↑	0.004	-	0.141	↓ *	-	↑	-	0.038
14th	PS 22:4/PGE1	↑	0.008	-	0.142	↓ *	-	↑	-	0.033
17th	3.20 901.5467m/z	↑	0.003	↓	0.026	-	-	-	-	0.075
19th	PE 18:1;O/20:0	↑	0.002	-	0.201	↓ *	-	↑	-	0.044
20th	PE 18:1;O2/18:0	↑	0.004	-	0.191	↓ *	-	↑	-	0.049
22nd	PE 18:1;O/18:0	↑	0.006	-	0.146	-	-	-	-	0.052
24th	PC 18:1;O/16:0	↑	0.005	-	0.186	-	-	-	-	0.063
1st	PC 36:4;O2	↓	0.009	-	0.098	-	-	-	-	0.054
2nd	PC 20:4;O/16:0	↓	0.009	-	0.116	↑ *	-	↓	-	0.042
3rd	PE 20:3;O/18:0	↓	0.004	-	0.133	↑ *	-	↓	-	0.016
8th	PG 40:5;O2	↓	0.002	↑	0.031	↑ *	↑ *	↓	↑ *	0.040
11th	PE 20:4;O/18:0	↓	0.008	-	0.135	↑ *	-	↓	-	0.022
13th	PC 16:0/20:4;O	↓	0.011	-	0.133	↑ *	-	↓	-	0.048
15th	PC 18:1;O/18:0	↓	0.004	-	0.214	-	-	-	-	0.052
16th	PC 18:1;O/18:0	↓	0.004	-	0.184	-	-	-	-	0.059
18th	PC 20:4;O/18:0	↓	0.005	-	0.134	↑ *	-	↓	-	0.029
21st	PC 18:1;O/16:0	↓	0.014	-	0.229	↑ *	-	↓	-	0.045
23rd	PC 18:1;O/16:0	↓	0.004	-	0.164	-	-	-	-	0.059
25th	Prostaglandin D2	↓	0.001	↑	0.022	↑ *	-	↓	-	0.008

* Significantly different from Obese + Low group; ↑ Increased; ↓ Decreased; - No significant change; PC: Phosphatidylcholine, PE: Phosphatidylethanolamine; PS: Phosphatidylserine; PG Phosphatidylglycerol; PI: Phosphatidylinositol. Statistical significance is displayed in red (*p* < 0.05).

## Data Availability

The original contributions presented in the study are included in the article/Appendix A, further inquiries can be directed to the corresponding author/s.

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
