# Peer review of "Combined Association of Plasma Metabolites with Body Mass Index and Physical Activity Level"

_biology, 2024, doi:10.3390/biology13121074_

Round 1

Reviewer 1 Report

Comments and Suggestions for Authors

This manuscript examines the relationship between plasma metabolites and body mass index in association with physical activity. A lot of similar studies are carried out every year. However, the value of this study lies in the study of the Brazilian ethnic group as well as the new interesting data on oxidized forms of lipids.

However, during the review of the article, serious comments were found that required a serious revision of the text of the article.

Article

The title of the article needs to be reconsidered because the word "modulation" does not reflect the meaning of the scientific work. It is better to write "correlation" or "relationship".

Abstract

The abstract should include the main findings, including metabolites that have been shown to correlate with BMI and physical activity.

Introduction

The Introduction is very short. The authors used only one third of the references in this chapter. This is a very strange decision.

Methods

The Methods are well described.

Results

There are elements in Figures 1 and 3 that are difficult to read. The text size needs to be increased. Key metabolites should be listed and the main correlations obtained should be described.

Discussion

The Discussion chapter contained twice as many references as the Introduction. A true discussion of the results obtained takes a couple of paragraphs. Constant comparison with scientific literature data obscures the true results of the article.

I propose to move most of the Discussion chapter to the Introduction, and in Chapter 4 itself to describe in detail my own conclusions obtained as a result of the experiments.

I also strongly recommend that authors structure the Discussion chapter. To do this, all metabolites can be divided into groups, each of which can be briefly discussed in this chapter.

Conclusions

The Conclusions is very short. It is necessary to indicate the key results of the article and draw general conclusions based on them.

Reviewer 2 Report

Comments and Suggestions for Authors

In the manuscript of M lambert et al., the authors address the question, how plasma metabolite levels measured by mass spectrometry correlate with Body Mass Index (BMI) and intensity of physical activity (Index of Physical Activity Questionnaire, IPAQ). The study groups included the subjects with Low/High BMI (Lean/Overweight-Obese), among which Low/High IPAQ groups were distinguished. Metabolite levels in Low BMI/Low IPAQ, Low BMI/High IPAQ, High BMI/Low IPAQ, High BMI/High IPAQ combinations were also analyzed.

In a large dataset, describing metabolite levels, 64 substances were identified whose levels had differences between the analyzed groups. The authors' correlation network and eigenvector centrality calculation identified 25 metabolites with the highest value of this parameter. Among these metabolites, the authors identified separate group of compounds whose changes in levels were associated with high BMI and low IPAQ, and another group of substances found in individuals with low BMI and high levels of physical activity. The directionality of the changes in the metabolic profile characteristic of the combination of low BMI and high IPAQ was opposite to those observed in high BMI subjects.

Overall, the peer-reviewed paper gives the impression of a well-designed and carefully conducted study with specific, interesting results.

The reviewer has no fundamental comments on the paper.

Minor comments.

1. Line 109: Whether IPAQ Questionnaires is reflected by reference # 19? A reference that specifically reflects the essence of the IPAQ is desirable.

2. Supplementary: It is highly desirable to add a brief commentary to the data description in the appendix (Tables S1-S3), as well as a deciphering of the labelled substances (PC, PE, etc.), and the parameters analyzed (F.val, E.S., etc.), to make this part of the paper easier to understand by a reader.

Reviewer 3 Report

Comments and Suggestions for Authors

The topic studied by the authors is of great scientific and practical importance for understanding normal metabolic processes and their pathological changes with increasing body weight and insufficient physical activity. This is topic number 1 in modern dietetics and preventive medicine.

The peer-reviewed study was carried out at a high scientific level; the authors used modern methods of blood analysis and the most advanced methods of statistical analysis. All this made it possible to obtain a result that will definitely interest readers and will significantly speed up further scientific research in this area.

Recommendations for improving the manuscript:

1. Correlation coefficients in Table 2 and in the text should be rounded to thousandths.

2. It is better to delete the first two columns of Table 2 (write this figure as text in the title of the table or in a note under it).

3. In Figure 3 you need to place trivial names of substances that are understandable to readers. These same names should be added as a second column in Table S1. Table S1 partially needs to be placed in the main text of the article immediately after Figure 3. Since this table contains so many columns, I recommend leaving the first 4 columns and these three columns “BMI (p), IPAQ (p), Interaction (p)” in it. . The full version of Table S1 remains in the supplementary materials.

4. In Figure 4, you need to remove the inscription above the picture. On the y-axis, tenths are separated by a dot, not a comma. On the x-axis you need to write trivial names that are understandable to readers (the same applies to the description of the material in the text of the article).

5. The rows in Table 3 must be arranged in order according to the “Obese” column: first with the most pronounced effect “↑ Increased”, and then with the increasing effect “↓ Decreased”. It is better to keep the standard vertical page layout for this table (it may be wise to discard a few of the least significant columns that convey little information).

6. Since not all readers can easily remember the formulas of each substance, I recommend adding a drawing to the discussion that will take up a whole page. In this figure, I advise you to depict the formulas of the most significant metabolites analyzed in this article, dividing them into several groups: first of all, “Obese”, “↑ Increased” and “↓ Decreased”. This drawing will probably become very popular on the Internet in the future. Its simplified version is best presented in the form of a graphic abstract of the article. This will significantly increase the number of citations for this article.

7. In magazine titles, most words must be capitalized (except for conjunctions, prepositions and articles). Article titles do not need to capitalize all words.
